# Pseudolaric Acid B Induces Growth Inhibition and Caspase-Dependent Apoptosis on Head and Neck Cancer Cell lines through Death Receptor 5

**DOI:** 10.3390/molecules24203715

**Published:** 2019-10-16

**Authors:** Su-Jung Choi, Chi-Hyun Ahn, In-Hyoung Yang, Bohwan Jin, Won Woo Lee, Ji-Hoon Kim, Min-Hye Ahn, Neeti Swarup, Kyoung-Ok Hong, Ji-Ae Shin, Nam-Tae Woo, Seong Doo Hong, Jae-Il Lee, Sung-Dae Cho

**Affiliations:** 1Department of Oral Pathology, School of Dentistry and Dental Research Institute, Seoul National University, Seoul 03080, Korea; anna47408@snu.ac.kr (S.-J.C.); chihyun610@snu.ac.kr (C.-H.A.); o117653266@snu.ac.kr (J.-H.K.); mine17@snu.ac.kr (M.-H.A.); neeti@snu.ac.kr (N.S.); hongko95@snu.ac.kr (K.-O.H.); sky21sm@snu.ac.kr (J.-A.S.); hongsd@snu.ac.kr (S.D.H.); jilee@snu.ac.kr (J.-I.L.); 2Cancer Center, School of Medicine, Texas Tech University Health Sciences Center, Lubbock, TX 79430, USA; inhyoung3@naver.com; 3Laboratory Animal Center, CHA University, CHA Biocomplex, Sampyeong-dong, Seongnam 13488, Korea; jnbhwan@cha.ac.kr (B.J.); lsw232@chamc.co.kr (W.W.L.); 4Department of ChemBio Specialization Program (CK-II), Dankook University, Cheonan 31116, Korea; natwoo2@gmail.com

**Keywords:** pseudolaric acid B, oral cancer, death receptor 5, caspase-8 Apoptosis

## Abstract

Pseudolaric Acid B (PAB), diterpenoid isolated from the root bark of *Pseudolarix kaempferi* Gordon tree (Pinaceae), exhibits an anti-proliferative and apoptotic activity in various cancer cell lines but to date, the effects of PAB on head and neck cancer (HNC) cell lines remain to be elucidated. In this study, we showed that PAB significantly inhibited the viability and caspase-dependent apoptosis in HN22 cell line. PAB-induced apoptosis is through inducing death receptor 5 (DR5) together with the increase in the expression of cleaved caspase-8. It also inhibited the proliferations and induced apoptosis through DR5 in other three HNC cell lines (HSC3, Ca9.22, and HSC4). Extending our in vitro findings, we found that ethanol extract of *Pseudolarix kaempferi* (2.5 mg/kg/day) reduced tumor growth in a xenograft model bearing HN22 cell line without any change in body weight. DR5 were also found to be increased in tumors tissue of PAB-treated mice without any apparent histopathological changes in liver or kidney tissues. Taken together, PAB may be a potential lead compound for chemotherapeutic agents against head and neck cancer.

## 1. Introduction

Head and neck cancers (HNCs) are the 15 most common cancers worldwide, with 500,550 incident cases in 2018, and the total number of deaths due to cancer of the lip and oral cavity was 67% of male deaths in 2018 [1]. The overall 5-year survival rate for people with lip and oral cavity cancers is less than 50% [2]. For early-stage lesions, surgical resection has been used to the treatment of HNCs. A variety of chemotherapeutic agents such as cisplatin, carboplatin, 5-fluorouracil, and taxanes are commonly used for co-treatment of HNC. Despite significant refinement of surgical and adjuvant treatment modalities, the prognosis for patients with HNC remains still poor and a 5-year survival rate has not changed significantly for several decades, which can be mostly by virtue of repeated local recurrences or distant metastasis [3]. Therefore, developing novel effective chemotherapeutic strategy is necessary.

From 1981 to 2014, many of the approved chemotherapeutic drugs were naturally originated [4]. Novel natural product derivatives play important roles in previously unresponsive malignancies due to their high efficacies and low unintended side effects, so development of natural product-based drug still has considerable utility in the demanding era of personalized cancer therapy [5]. Pseudolaric acid B (PAB), a novel diterpene acid, is processed from dried cortex of the roots of *Pseudolarix kaempferi* [6]. Since the 17th century, PAB has been used for the fungal skin infection in China and its antifungal activity was reported in the scientific literature for the first time in 1957 [7]. Up to 2019, there have been over 100 publications relevant to different aspects of researches on PAB such as anti-virus, anti-angiogenic, anti-fertility etc. [7]. Among the studies, it has shown that PAB has strong cytotoxic effects on various types of cancers, including colorectal cancer, gastric cancer, liver cancer, prostate cancer, lung cancer, melanoma, and leukemia [8,9,10,11,12,13,14,15,16]. However, there has not been yet a single research on HNCs. We for the first time scrutinize the efficacy and associated molecular targets of PAB on HNC cell lines in culture and an animal model.

## 2. Results

### 2.1. Effect of PAB on the Viability of HN22 Human HNC Cell Line

To determine the anti-proliferative activity of PAB, HN22 cells were treated with PAB at various concentrations. IC_50_ value of PAB was found to be approximately 0.7 µm/mL at 24 h, so 0.5 and 1µm was been used in this next experiment. First, the viability of HN22 cells treated with PAB were measured using a trypan blue exclusion assay. PAB caused a significant decrease in HN22 cell viability in a concentration-dependent manner (Figure 1B). PAB-treated HN22 cells were also stained using the live/dead assay kit. The dead cells (red stained) were significantly increased after the PAB treatment (Figure 1C). From these striking results, we suggest that PAB can decrease cell viability in HNC cell line.

### 2.2. Effect of PAB on Caspase-Dependent Apoptosis in HN22 Human HNC Cell Line

To evaluate whether the observed growth inhibition and cell death are related to apoptosis, we detected cleaved caspase-3 and cleaved PARP, hallmarks of apoptosis to evaluate the apoptotic effect of PAB in HN22 cells. The cleaved form of two proteins were clearly observed by PAB in a concentration-dependent manner indicating that PAB activates the apoptotic pathway (Figure 2A). Z-VAD, a general caspase inhibitor that irreversibly binds to the catalytic site of caspase proteases, was used to verify the role of caspase activation in the induction of apoptosis. Z-VAD noticeably blocked apoptosis caused by PAB in HN22 cells (Figure 2B). We confirmed PAB induced apoptosis by assessing flow cytometry experiments. The number of annexin V-positive cells in PAB treatment group was increased more than control treatment group (Figure 2C). These results provide the evidence that anti-proliferative and cytotoxic effects of PAB are as a consequence of caspase-dependent apoptosis in HN22 human HNC cell line.

### 2.3. Involvement of DR5 in PAB-Induced Apoptosis in HN22 Human HNC Cell Line

Although no study has indicated that direct involvement of PAB in the expression of the DRs, our previous studies have demonstrated that many naturally derived products have direct involvement in the expression of the DR5 [17,18,19,20]. To explore the potential molecular targets by which PAB induces apoptosis in HN22 cells, we performed RT-PCR. We found that the level of DR5 mRNA was remarkedly increased by PAB (Figure 3A,B) which means it was transcriptionally regulated by PAB. We also performed Western blotting and immunofluorescence staining for the detection of DR5 protein. The results were consistent with our putative interactions from previous studies (Figure 3C–E). Since caspase-8 activation is critical for extrinsic apoptosis pathway via DR5, we detected the effect of PAB on caspase-8 activation. The results showed that PAB increased the expression level of cleaved caspase-8 (Figure 3F). These results suggest that DR5 may be involved in PAB-induced apoptosis in human HNC cells.

### 2.4. Growth-Inhibitory and Apoptotic Effects of PAB via DR5 Human HNC Cell Lines

To generalize the growth-inhibitory and apoptotic effects of PAB in human HNC, we used three other human HNC cell lines (HSC-3, Ca9.22, and HSC-4). The results showed that PAB significantly suppressed viability of three other HNC cell lines (Figure 4A) and it led to dramatic increases in the expression levels of DR5, cleaved PARP and cleaved caspase-3 (Figure 4B). To confirm PAB-induced apoptosis in HSC-3, Ca9.22, HSC-4, we performed annexin V staining. The number of annexin-positive cells was much higher in PAB-treated those cell lines than in control group (Figure 4C). These results suggest that apoptosis may represent a general mechanism for the anti-cancer effect of PAB in human HNC cell lines.

### 2.5. Anti-Tumorigenic Effect of EEPK in a Tumor Xenograft Model Bearing HN22 Cells

Based on the above cell culture results showing strong biological activities of PAB in HNC cell lines, we investigate the effect of EEPK, which includes PAB, on the growth of HN22 cell xenograft in Balb/c nu/nu male mice. HN22 cells were subcutaneously injected into mice and the mice were intraperitoneally injected with vehicle control or EEPK (2.5 mg/kg/day) (*n* = 6 per group) for 25 day. Throughout the study, we observed that EEPK significantly suppressed tumor volume of HN22 cell xenograft and at the day 25 (the end of study), EEPK showed 40% (*p* = 0.037) reduction in tumor volume in comparison to control group (Figure 5A,B). EEPK also showed a strong trend toward significance of a decrease in tumor weight (Figure 5C). The tumors were isolated and homogenized to measure the protein levels of DR5 and cleaved caspase-3. The results showed that EEPK dramatically increased both DR5 and cleaved caspase-3 (Figure 5D). To test whether EEPK could be an effective strategy to reduce in vivo tumor growth without any toxicity, we measured body and organ weight (liver and kidney). The result showed that either body or organ weight was not changed by EEPK (Figure 5E,F). In addition, there was no difference in histopathological findings between control and EEPK treatment group (Figure 5G). Collectively, these results suggest that EEPK has an anti-tumorigenic effect without hepatic and nephrotic toxicity.

## 3. Discussion

Although many researchers made various attempts to find a novel anti-HNC strategy for HNC treatments, the survival rates of HNC and the treatment of this deadly disease have not been improved yet. In this study, we demonstrated that PAB has in vitro anti-cancer effects including anti-proliferative and caspase-dependent apoptotic activities in HNC cell lines and associated with DR5 protein indicating that the extrinsic pathway may play an important role in PAB-induced apoptosis of HNC. Moreover, in vivo test using a tumor xenograft model bearing HN22 cells demonstrated that remarkable anti-tumor activity of EEPK without any side effects on body and organ weights and histopathological findings of liver and kidney. These results suggest that the apoptotic activity of PAB targeting DR5 may serve as an attractive apoptosis inducer for HNC treatments.

Many researchers have shown that natural compounds have been used as sources for the development of anti-cancer drugs [21,22]. From our previous studies, we reported that various natural agents such as nitidine chloride, caffeic acid phenethyl ester, and oridonin have very distinctive anti-cancer effects in HNC cell lines [23,24,25]. It has been also demonstrated the possibility of PAB as a promising phytochemical anticancer agent against human cancers. Several studies found that PAB induced apoptosis and inhibited angiogenesis by interacting with a binding site on tubulin like other microtubule-targeting agents, such as paclitaxel [26,27]. PAB also induced cell senescence and apoptosis in various cancer cell lines by activation of p53 and c-Jun N-terminal kinase which are associated with induction of apoptosis [28,29,30,31]. These suggest that PAB can be a strong apoptosis-inducing agent against various cancers just like our present results. Although numerous studies demonstrated caspase-dependent apoptosis, one previous study by Khan et al. [16] showed that PAB partially induced caspase-independent apoptosis in U87 Glioblastoma cells through apoptosis inducing factor. In the present study, we examined the caspase dependency of PAB-induced apoptosis using a pan-caspase inhibitor, z-VAD. The results showed that z-VAD partially blocked PAB-induced apoptosis in HN22 cells (Figure 2B). The reason may be because z-VAD did not completely inhibit caspase-3 activation. However, we still think that we should not rule out the possibility that PAB induces caspase-independent apoptosis in HNC cell lines.

To respond the stress such as heat, radiation, and nutritional deprivation, DR5 recruits Fas-associated death domain (FADD), TNFR1-associated death domain, and pro-caspase-8 to form death-inducing signaling complex, which trigger apoptosis. Thus, DR5 is a critical mediator for the apoptotic pathway and targeting DR5 has high successful potential as a novel chemotherapeutic strategy. DR5 also represents one of the most important extrinsic pathways activated by naturally derived anti-cancer agents [32,33]. Previously, our group also reported that natural product extracts or naturally derived single compounds exhibited anti-cancer effects through the induction of DR5 in either in vitro or in vivo HNC experimental models [19,34,35,36] indicating that DR5 may be good candidate target to induce apoptosis in HNC. However, most of findings on PAB-induced apoptosis focused on intrinsic apoptotic pathway. For examples, PAB inhibits mitochondria-mediated apoptosis pathways in human gastric carcinoma BGC-823 and MKN-45 cells and cervical cancer HeLa cells through the modification of the ratio of Bax/Bcl-2 [12,37]. PAB also induced apoptosis via proteasome-mediated protein degradation of Bcl-2 in human prostate cancer DU145 cells [38]. Yu et al. showed that the PAB-mediated apoptosis in human breast cancer MCF-7 cells was independent of the death receptor pathway evidenced by no change of the expression levels of FADD and Fas L, which are related to the extrinsic pathway [39]. These indicate that PAB-mediated apoptosis in human cancer cell lines may be mainly associated with the intrinsic pathway. In the present study, we investigated the effects of PAB on the expression levels of Bcl-2 and Bax protein. Unfortunately, our unpublished data showed that PAB did not affect the expression of those proteins (data not shown). However, PAB clearly increased the expression of DR5 protein in a transcriptional modification (Figure 3) suggesting that PAB-induced apoptosis in HNC cell lines favors extrinsic pathway over intrinsic one. To the best of our knowledge, this is for the first time to confirm the direct involvement of PAB in the expression of DR5 during apoptosis. Nevertheless, more detailed mechanisms should be evaluated to determine how DR5 protein is transcriptionally regulated by PAB in HNC cell lines in the future.

Previously, several studies investigated the in vivo toxic effect of PAB. PAB significantly suppressed the tumor growth in nude mice at a dose of 15 mg/kg and 25 mg/kg, but any sign of toxicity or body weight loss appeared [26]. In addition, using Kunming mice, 25 mg/kg dose of PAB did not cause any detectable toxic effect in live and kidneys [16]. In the present study, we also found that EEPK (2.5 mg/kg) dramatically suppressed tumor growth, whereas control tumors grew rapidly and associated with DR5 expression (Figure 5A–D), consistent with our in vitro results. Moreover, EEPK did not exhibit obvious treatment-related toxicity, such as the loss of body and organ weights, and abnormal histopathological findings of liver and kidney (Figure 5E–G). Although a detailed study for the toxicity of PAB still needs to be performed, EEPK can have anti-tumor activities in the tumor xenograft model bearing HN22 cells without any toxicity.

Collectively, this study reveals induction of DR5-mediated extrinsic apoptosis for anti-HNC therapy using PAB. It is worth to say PAB may be a potential lead compound for developing future chemotherapeutic agents against HNC.

## 4. Materials and Methods

### 4.1. Cell Culture

HN22 HNC cell line was kindly obtained from Prof. Lee of Dankook University (Cheonan, Korea). HSC-3, HSC-4, and Ca9.22 human HNC cell lines were provided from Prof. Shindoh of Hokkaido University (Hokkaido, Japan) and each cell lines were grown in DMEM supplemented with 10% fetal bovine serum (FBS) and antibiotics (Penicillin-streptomycin) at 37 °C in 5% CO_2_ incubator. All experiments were prepared in cells cultured at 50–60% confluence.

### 4.2. Subsection Chemicals and Antibodies

PAB was purchased commercially from Sigma-Aldrich (St Louis, MO, USA) and its chemical structure is seen in Figure 1A. It was dissolved in dimethyl sulfoxide (DMSO), aliquoted, and stored at −20 °C. Final concentration of DMSO did not exceed 0.1%. Ethanol extract of *Pseudolarix kaempferi* (EEPK), which includes PAB, a main active component, was kindly provided by Dr. Woo. Annexin V-FITC/PI was supplied by BD Biosciences (Franklin Lakes, NJ, USA). Antibodies against cleaved poly (ADP-ribose) polymerase (PARP), cleaved caspase-3, and death receptor 5 (DR5) were supplied by Cell Signaling Technology, Inc. (Charlottesville, VA, USA). Antibody against β-Actin was obtained from Santa Cruz Biotechnology, Inc. (Dallas, TX, USA). Z-VAD was purchased from R&D systems (Minneapolis, Minnesota, USA).

### 4.3. Trypan Blue Exclusion Assay

Cells were seeded in 6-well plates and incubated with various concentrations of PAB for 24 h. Cells were stained with 0.4% trypan blue (Gibco, Paisley, UK), and viable cells were then counted using a hemocytometer. All experiments were performed independently three times with triplicate samples in each experiment.

### 4.4. Live/Dead Assay

Cytotoxic effect of PAB was evaluated by Live/Dead Viability/Cytotoxicity assay (Life Technologies, Grand Island, NY, USA). The polyanionic dye Calcein-AM is retained within live cells, producing an intense green fluorescence through intracellular esterase activity. Ethidium homodimer-1 enters cells with damaged membranes and binds to nucleic acids, producing a bright red fluorescence in dead cells. Briefly, cells were stained with 2 μm Calcein-AM and 4 μM Ethidium homodimer-1, and then incubated for 30 min at room temperature (RT). Cells were analyzed under a fluorescence microscopy.

### 4.5. Western Blot Analysis

Whole-cell lysates were extracted with RIPA lysis buffer and protein concentration in each sample was measured using a DC Protein Assay Kit (BIO-RAD Laboratories, Madison, WI, USA). After normalization, equal amounts of protein were separated by sodium dodecyl sulfate polyacrylamide gel electrophoresis and then transferred to Immuno-Blot PVDF membranes. The membranes were blocked with 5% skim milk in tris-buffered saline with Tween 20 (TBST) at RT for 2 h and incubated with primary antibodies and corresponding horseradish peroxidase-conjugated secondary antibodies. The immunoreactive bands were visualized by ImageQuantTM LAS 500 (GE Healthcare Life Sciences, Piscataway, NJ, USA).

### 4.6. Annexin V/PI Double Staining

The induction of apoptosis was measured using an FITC Annexin V Apoptosis Detection Kit (BD Pharmingen, San Jose, CA, USA) according to the manufacturer’s protocol. Briefly, floating and adherent cells were collected, washed twice with phosphate buffered saline (PBS), and pelleted by centrifugation. Cells were resuspended in annexin V binding buffer containing 3 μL annexin V-FITC and 1 μL PI and incubated for 15 min at RT in the dark. Subsequently, cells were transferred to a FACS tube and analyzed by flow cytometry using a FACS Calibur (BD Biosciences, San Jose, CA, USA), and at least 10,000 events were counted per sample.

### 4.7. Quantitative Real-Time PCR

Total RNA was extracted using Trizol Reagent (Life Technologies, Carlsbad, CA, USA). One microgram of RNA was reverse-transcribed by an AMPIGENE cDNA Synthesis Kit (Enzo Life Sciences, Inc., NY, USA), and the resultant cDNA was subjected to PCR using AMPIGENE qPCR Green Mix Hi-Rox (Enzo Life Sciences, Inc.). Real-time PCR was performed using the Applied Biosystems StepOne Plus Real-Time PCR System (Applied Biosystems, Foster city, CA, USA) and PCR conditions for all genes were as follows: 95 °C for 2 min, followed by 40 cycles of 95 °C for 10 sec and 60 °C for 30 s. The relative amount of each gene was normalized to the amount of GAPDH and calculated using the 2^−ΔΔC^_T_ method. The qPCR primers were: DR5 sense, 5′-CACCTTGTACACGATGCTGATA-3′, DR5 anti-sense, 5′-CTCAACAAGTGGTCCTCAATCT-3′, GAPDH sense, 5′-GTGGTCTCCTCTGACTTCAAC-3′, GAPDH anti-sense, 5′-CCTGTTGCTGTAGCCAAATTC-3′.

### 4.8. Reverse Transcription-Polymerase Chain Reaction

Total RNA was extracted from easy-BLUE Total RNA Extraction Kit (iNtRON, Daejeon, Korea), and then 1 μg of total RNA was transcribed into cDNA using TOPscript RT DryMIX (Elpis Biotech, Daejeon, Korea). cDNA was subjected to PCR using HiPi PCR PreMix (Elpis Biotech, Daejeon, Korea). DR5 and β-Actin transcripts were amplified by PCR using the following specific primer; DR5 sense 5′-ATGAGATCGTGAGTATCTTGCAGC-3′ and DR5 anti-sense 5′-TGACCCACTTTATCA GCATCGTGT-3′, GAPDH sense 5′-ACCAGGGCTGCTTTAACTC-3′ and GAPDH anti-sense 5′-GCTCCCCCCTGCAAATGA-3′. DR5 amplification was performed in 30 cycles (1 min at 95 °C, 1 min at 57.8 °C, 1 min 30 s at 72 °C) and β-Actin amplification was performed in 25 cycles (1 min at 95°C, 1 min at 60°C, 1 min 30s min at 72 °C). The PCR products were separated by electrophoresis on a 1.2% agarose gel and visualized with ethidium bromide.

### 4.9. Immunofluorescence Staining

HN22 cells were seeded on 4-well plates and treated with PAB. After 24 h, cells were fixed and permeabilized using the Cytofix/Cytoperm solution for 1 h at 4 °C. Cells were blocked with 1% bovine serum albumin in PBS 1 h at RT and incubated overnight at 4 °C with antibodies against DR5 followed by incubation with FITC-conjugated secondary antibody for 1 h at RT. Cells were observed using a fluorescence microscopy with the appropriate filters for DAPI or FITC dyes.

### 4.10. Tumor Xenograft Model

Six-week-old Balb/c nu/nu male mice (NARA Biotech, Pyeongtaek, Korea) were caged in a facility with a 12 h light/dark cycle and allowed Teklad diet (2018s) and water ad libitum. All mice were handled according to Institutional Animal Care and Use Committee (IACUC) guidelines approved by CHA university (IACUC approval number: 190125). HN22 cells were subcutaneously injected into the flanks of the mice, and then, the mice were assigned randomly to two treatment groups (*n* = 6 for each group). About 10 days after incubation, tumor-bearing mice received vehicle control (phosphate buffered saline) or 2.5 mg/kg/day of EEPK (*i.p.*) five times per week for 25 days. Tumor volume and body weight were monitored twice a week. The mice were sacrificed for examination of tumor and organ weight.

### 4.11. Histopathological Examination of Organs

Mice organs (liver and kidney) were fixed in 10% neutral buffered formalin. Tissue sections were cut at a thickness of 5 μm and stained with hematoxylin and eosin. Histopathological changes were analyzed using microscope equipped with DFC550 digital camera (Leica, Wetzlar, Germany).

### 4.12. Statistical Analysis

The data analyzed with GRAPHPAD PRISM V software (La Jolla, CA, USA). Statistical analyses were performed using SPSS version 22 (SPSS Inc., Chicago, IL, USA). Two-tailed Student’s t-tests for two comparisons and one-way ANOVA with Tukey’s post hoc test for multiple comparisons were used. Statistical evaluation for in vivo study was calculated by Mann-Whitney test in SPSS. Values of *p* < 0.05 were considered significant (*).

## Figures and Tables

**Figure 1 molecules-24-03715-f001:**
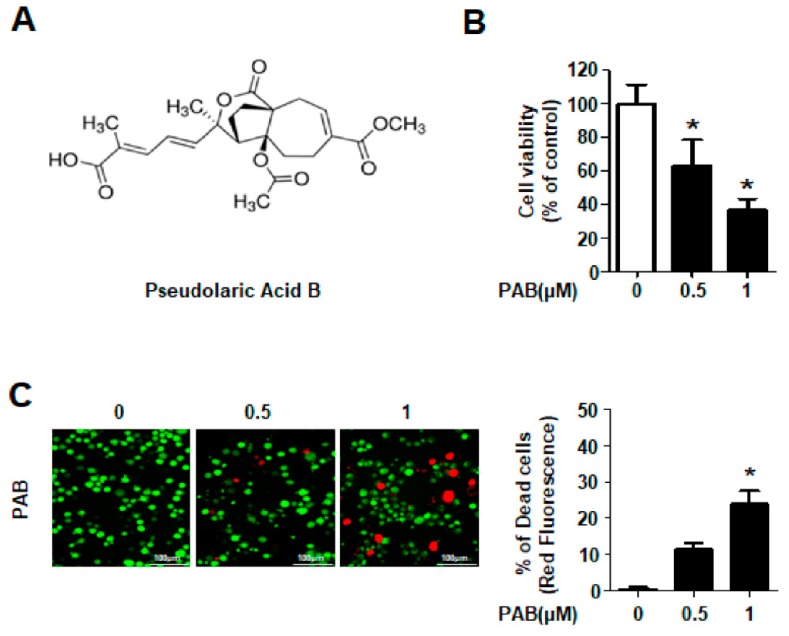
Effects of PAB on the survival of HN22 cell line. (**A**) The chemical structure of PAB. (**B**) HN22 cells were treated with DMSO or designated concentrations (0.5 and 1 μm) of PAB for 24 h. Cells viability was examined by a trypan blue exclusion assay. (**C**) Live (green) and dead (red) cells were determined by Live/dead assay kit as mentioned in ‘Materials and Methods’ (magnification, X200). Columns represent means ± SD of triplicate determinations. (* *p* < 0.05).

**Figure 2 molecules-24-03715-f002:**
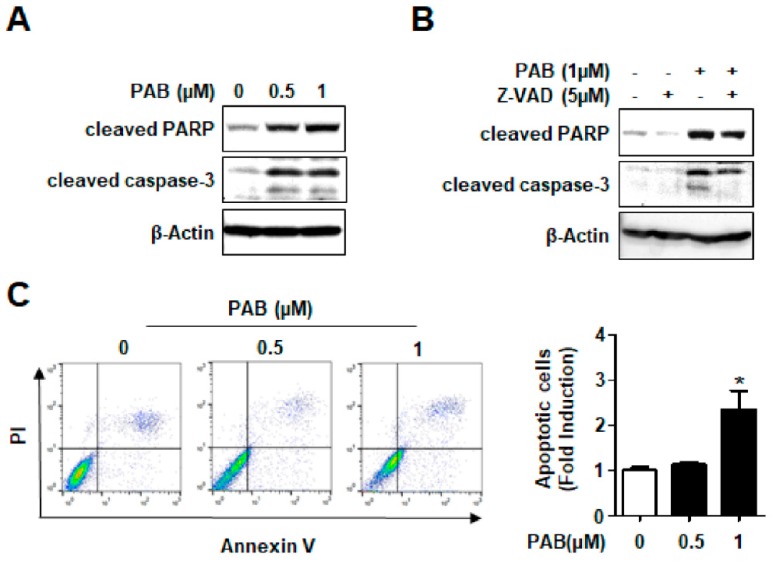
Effects of PAB on caspase-dependent apoptosis in HN22 cell line. (**A**) Western blotting using the antibodies against cleaved PARP and cleaved caspase-3 was performed. Actin was used as a loading control. (**B**) HN22 cells were pre-treated with 5 μM Z-VAD (a pan-caspase inhibitor) for 1hr prior to PAB. Protein level of cleaved PARP and cleaved caspase-3 were analyzed by western blotting. (**C**) Fluorescence-activated cell sorting (FACS) analysis of annexin V/ propidium iodide (PI) staining. The graph depicts the mean ± SD of triplicates (* *p* < 0.05).

**Figure 3 molecules-24-03715-f003:**
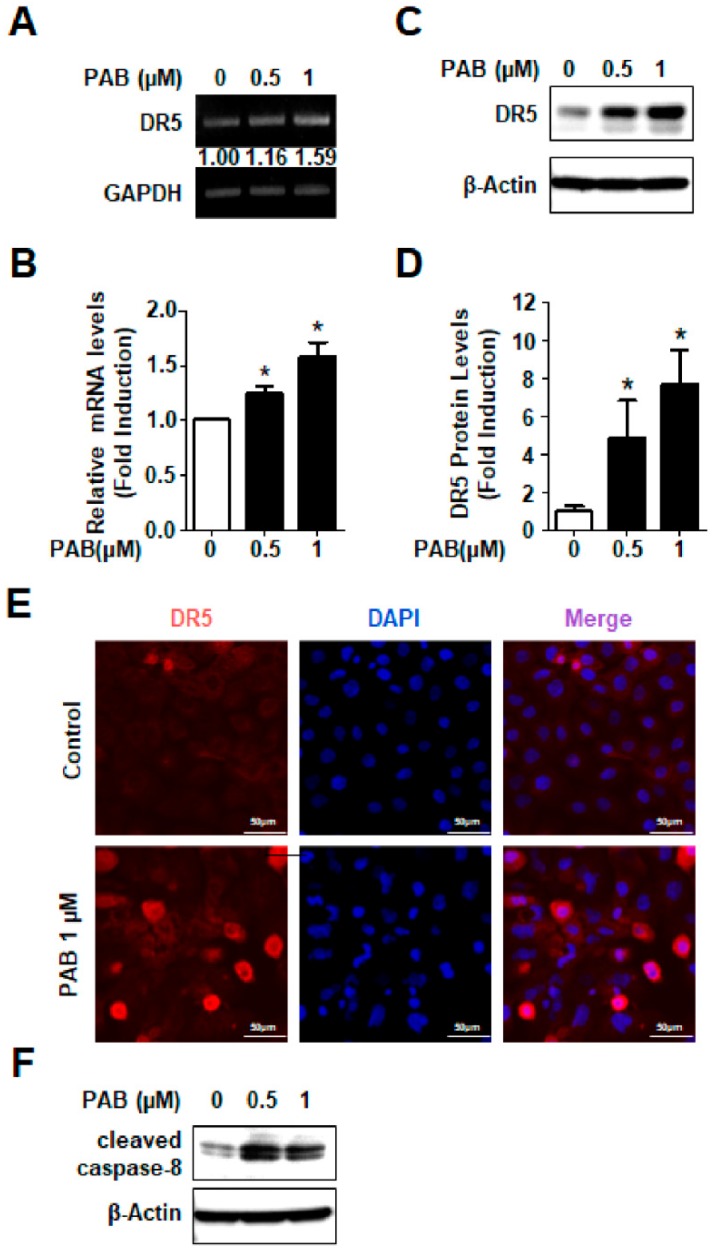
Involvement of DR5 in PAB-induced apoptosis in HN22 cell line. (**A**) DR5 mRNA was determined by RT-PCR. (**B**) mRNA levels were measured by QPCR. (**C**) HN22 cells were treated with DMSO or designated concentrations (0.5 and 1 μm) of PAB for 24 h as in subjected to DR5 protein and analyzed by western blotting. (**D**) The graph depicts the mean ± SD of triplicates. (* *p* < 0.05) (**E**) DR5 protein was detected by immunofluorescence staining (magnification, X400). (**F**) Cleaved caspase-8 was detected by western blotting.

**Figure 4 molecules-24-03715-f004:**
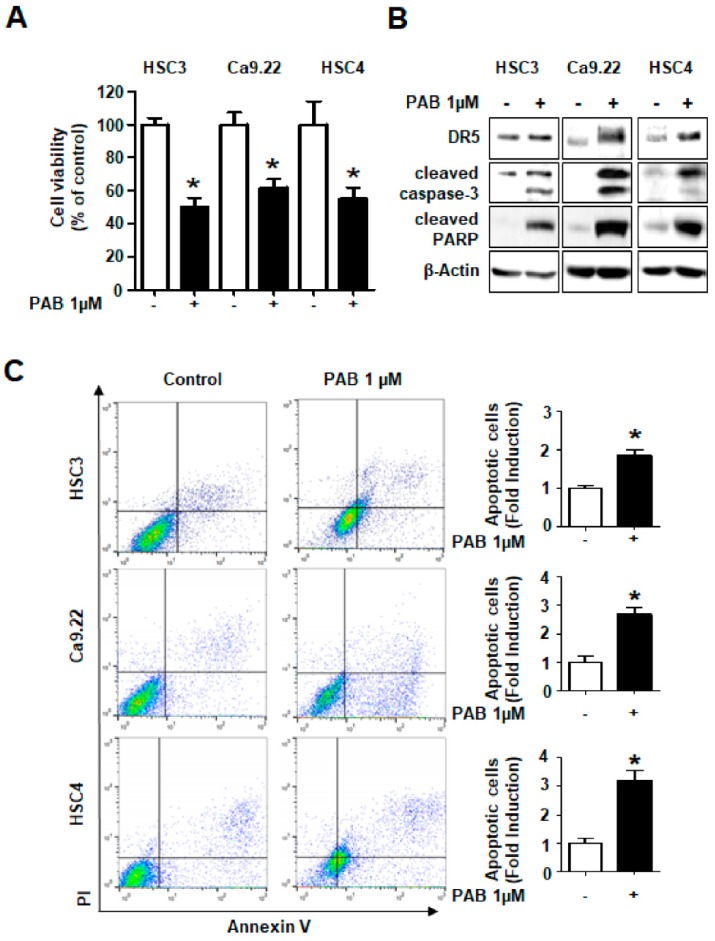
Growth-inhibitory and apoptotic effects of PAB in HSC3, Ca9.22, and HSC4 cells. (**A**) HSC3, Ca9.22, and HSC4 cells were treated with DMSO or PAB for 24 h, respectively. Effect of PAB on cell viability was examined by trypan blue exclusion assay. (**B**) Protein levels of DR5, cleaved caspase-3, and cleaved PARP were analyzed by western blotting. (**C**) Fluorescence-activated cell sorting (FACS) analysis of annexin V/ propidium iodide (PI) staining. The graphs depict the mean ± SD of triplicates. (* *p* < 0.05).

**Figure 5 molecules-24-03715-f005:**
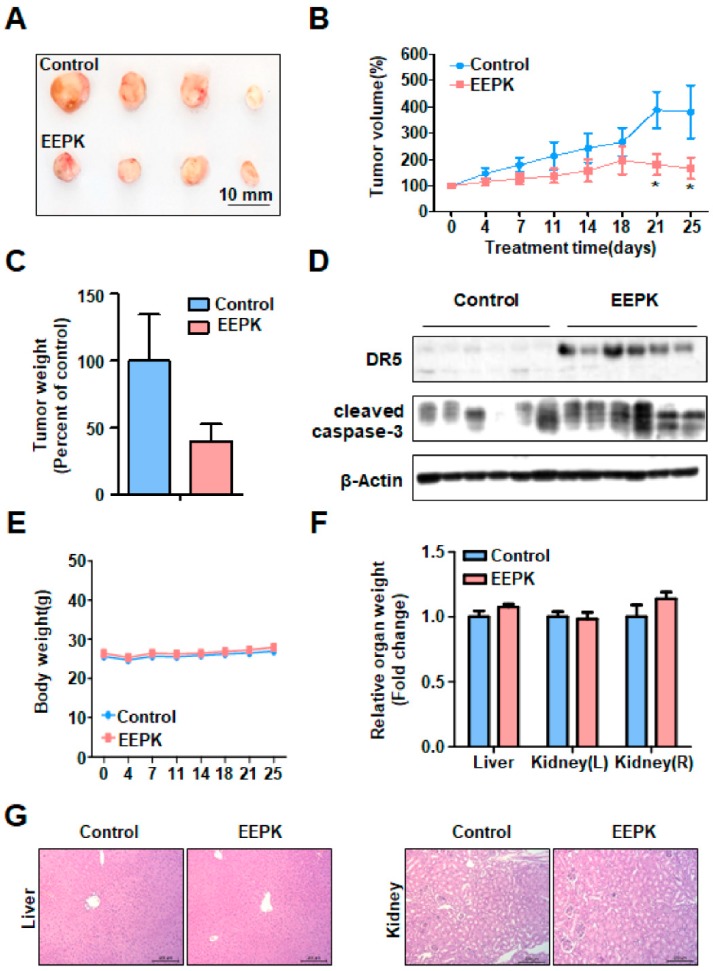
Effects of EEPK on tumor growth in a nude mouse xenograft model bearing HN22 cells. HN22 cells was inoculated with subcutaneous injection into the flanks of the mice, and mice were then received 2.5mg/kg/day of EEPK (i.p) five times per week for 25 day (*n* = 6 per group). (**A**) Representative images of tumors of mice after sacrificed. Tumor volume (**B**), tumor weight (**C**), of vehicle control or EEPK-treated groups were monitored as described in Materials and methods. Graphs represent mean ± SD and significance (*p*  < 0.05) compared to the vehicle control group is indicated (*). (**D**) Expression level of DR5 and cleaved caspase-3 detected by Western blotting. (**E**) Body weight were measured as described in Materials and methods. (**F**) Liver and kidney were surgically removed, and their weights were measured. Data are presented as the mean± S.E. (**G**) Histopathological changes of liver and kidney were visualized by H&E staining. Representative images are shown (magnification, X100).

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
