# Peer review of "Pseudolaric Acid B Induces Growth Inhibition and Caspase-Dependent Apoptosis on Head and Neck Cancer Cell lines through Death Receptor 5"

_molecules, 2019, doi:10.3390/molecules24203715_

Round 1

Reviewer 1 Report

Manuscript molecules-612773  disclosed the inhibitory effects of Pseudolaric Acid B (PAB) on head and neck cancer (HNC).  PAB was found to inhibit the proliferation and induced apoptosis through death receptor 5 (DR5) in several HNC cell lines. Later, the ethanol extract of Pseudolarix Kamperi was found to suppress tumor growth in vivo. The finding would be helpful to find a potential therapy for HNC. Thus, this paper is recommended to be published on Molecules after minor concerns.

1. Why did the authors use EEPK for in vivo study instead of PAB?

Author Response

Manuscript molecules-612773  disclosed the inhibitory effects of Pseudolaric Acid B (PAB) on head and neck cancer (HNC).  PAB was found to inhibit the proliferation and induced apoptosis through death receptor 5 (DR5) in several HNC cell lines. Later, the ethanol extract of Pseudolarix Kamperi was found to suppress tumor growth in vivo. The finding would be helpful to find a potential therapy for HNC. Thus, this paper is recommended to be published on Molecules after minor concerns.

Why did the authors use EEPKfor in vivo study instead of PAB?

à Your comment is totally right and we should conduct animal experiments with PAB, not EEPK. Frankly speaking, we couldn’t do nude mice xenograft model with PAB because PAB is too expensive (10mg of PAB is about 900 US dollars). Because of our research grant situation, we wanted to find an alternative way at that time. Recently, He et al. (Ref. 1) reported that the content of PAB was very high in Pseudolarix kaempferi extract. Thus, we used EEPK for the animal study.

Ref.1 He S, Li S, Yang J, Ye H, Zhong S, Song H, Zhang Y, Peng C, Peng A, Chen L Application of step-wise gradient high-performance counter-current chromatography for rapid preparative separation and purification of diterpene components from Pseudolarix kaempferi Gordon. J Chromatogr A. 2012 Apr 27;1235:34-8.

Reviewer 2 Report

Cho et al have presented an interesting article entitled as Pseudolaric Acid B Induces Growth Inhibition and Caspase-Dependent Apoptosis on Head and Neck Cancer Cell lines through Death Receptor.

This article will attract the attention of medicinal chemists. The following corrections are suggested for publication in Molecules.

The authors should proofread the manuscript and make it uniform  in vitro vs in

vitro 

(e.g., page 1, line 30; page 9, line 162) caspase 3 vs caspase-3. (page 3, line 79; page 6, line 119) 2. Pseudolarix kaempferi should be in italic (page 1, line 23, 24) 3. What is Pseudolarix Kamperi? Are Pseudolarix kaempferi and Pseudolarix Kamperi same? 4. Citations in the manuscript should be uniform and correct. See line 32 and 58 in page 2. 5. Line 64, page 2: value 6. Care should have been taken to write the structure of pseudolaric acid B. Revise the structure with correct stereochemistry.

Author Response

Cho et al have presented an interesting article entitled as Pseudolaric Acid B Induces Growth Inhibition and Caspase-Dependent Apoptosis on Head and Neck Cancer Cell lines through Death Receptor.

This article will attract the attention of medicinal chemists. The following corrections are suggested for publication in Molecules.

The authors should proofread the manuscript and make it uniform  in vitro vs invitro 

(e.g., page 1, line 30; page 9, line 162) caspase 3 vs caspase-3. (page 3, line 79; page 6, line 119)

à We read our manuscript all carefully. As you suggested, we made them uniform.

Pseudolarix kaempferi should be in italic (page 1, line 23, 24)

à As you suggested, we did that Pseudolarix kaempferi should be in italic.

What is Pseudolarix Kamperi? Are Pseudolarix kaempferi and Pseudolarix Kamperi same?

à We corrected it

Citations in the manuscript should be uniform and correct. See line 32 and 58 in page 2.

à As you suggested, we updated and rearranged the citations in the manuscript [Page 2; lines 46, 51]

Line 64, page 2: value

à We corrected it. [Page 2; line 70]

Care should have been taken to write the structure of pseudolaric acid B. Revise the structure with correct stereochemistry.

à As you suggested, we checked the chemical structure of PAB using data sheet from Sigma Aldrich’s. We replaced it with new one. (Figure 1A)

Reviewer 3 Report

The authors present a study on the mechanism of action of psedolaric acid B (PAB) in head and neck cancers. In vitro and in vivo work is presented and a clear anticancer effect is shown.

The manuscript is well written and accessible to the reader.

Criticism:

In the discussion, it is stated that PAB binds to tubulin. It would be very helpful to compare the effect of PAB to a clinically established drug such as taxol (paclitaxel). Without this comparison the statement that PAB is a clinical candidate cannot be made. The authors have to show that PAB is at least as effective as taxol or has some advantage over it. In the introduction it is stated the ca. 50% of chemotherapeutics are natural products or their derivatives. The paper of Newman is cited, when that paper is read it is clear that Newman’s definition is VERY liberal towards natural products so this number of 50% is meaningless. I recommend that the statement is toned down reflecting the more realistic situation such as the authors do in the discussion lines 169 and 170.3. Discussion, lines 203-204: why is this data not presented? If there is no effect the statement “data not shown” can be used.

Small things:

Line 43: used for co-treatment

Line 58: punctuation after reference

Line 64: IC50 – 50 in subscript

Line 67: is Figure 1A mentioned earlier in the text?

I am happy to recommend this manuscript for publication in Molecules if the issues raised are addressed.

Author Response

The authors present a study on the mechanism of action of psedolaric acid B (PAB) in head and neck cancers. In vitro and in vivo work is presented and a clear anticancer effect is shown.

The manuscript is well written and accessible to the reader.

Criticism:

In the discussion, it is stated that PAB binds to tubulin. It would be very helpful to compare the effect of PAB to a clinically established drug such as taxol (paclitaxel). Without this comparison the statement that PAB is a clinical candidate cannot be made.

à We fully understood what your comment is. As we mentioned, even though PAB was previously reported to act as a microtubule-targeting agent like taxol, in this study we focused on the role of PAB on targeting DR5 protein. That is the reason why we did not compare both chemicals. However, your comment is really valuable for our study, so we a little bit toned down our statement like the below.

the apoptotic activity of PAB targeting DR5 may serve as an attractive drug candidate for HNC treatments. à the apoptotic activity of PAB targeting DR5 may serve as an attractive apoptosis inducer for HNC treatments [Page 9; lines 189]

Thank you for your valuable comments.

The authors have to show that PAB is at least as effective as taxol or has some advantage over it.

à Although both PAB and taxol are microtubule-targeting agents and microtubule binding site of PAB is not clear yet, other antimitotic drugs bind to many diverse sites on tubulin and microtubules. It means that the effects of PAB and taxol may be different. Thus, it is worthwhile to compare the effect of PAB with taxol in future study. Thank you for your kind suggestion.

In the introduction it is stated the ca. 50% of chemotherapeutics are natural products or their derivatives. The paper of Newman is cited, when that paper is read it is clear that Newman’s definition is VERY liberal towards natural products so this number of 50% is meaningless. I recommend that the statement is toned down reflecting the more realistic situation such as the authors do in the discussion lines 169 and 170.3.  

à At that time, we wanted to give the readers more accurate number for that. However, we also thought your comment is totally right. Thus, we a little bit toned down our statement like the below.

From 1981 to 2014, nearly 51% of the approved chemotherapeutic drugs were naturally originated

à From 1981 to 2014, many of the approved chemotherapeutic drugs were naturally originated [Page 2; line 53]

Discussion, lines 203-204: why is this data not presented? If there is no effect the statement “data not shown” can be used.  

à As you suggested, we used “data not shown” for our unpublished data that has no effect. [Page 10; lines 225-226]

Small things:

Line 43: used for co-treatment

à As you suggested, we changed it. [Page 2; line 48]

Line 58: punctuation after reference

à As you suggested, we changed it.

Line 64: IC50 – 50 in subscript

à As you suggested, we changed it. [Page 2; line 70]

Line 67: is Figure 1A mentioned earlier in the text?

à We mentioned Figure 1A in the “Materials and Methods” section [Page 10; line 257]

Round 2

Reviewer 3 Report

The authors explain in their answer why they do not do a direct comparison to taxol at this stage. I would like to see this explanation incorporated into the text of the manuscript for the benefit of our readers.

Otherwise, the manuscript is quite interesting.